# MiR-23b and miR-133 Cotarget TGFβ2/NOTCH1 in Sheep Dermal Fibroblasts, Affecting Hair Follicle Development

**DOI:** 10.3390/cells13060557

**Published:** 2024-03-21

**Authors:** Junmin He, Chen Wei, Xixia Huang, Guoping Zhang, Jingyi Mao, Xue Li, Cunming Yang, Wenjing Zhang, Kechuan Tian, Guifen Liu

**Affiliations:** 1Shandong Key Laboratory of Animal Disease Control and Breeding, Institute of Animal Science and Veterinary Medicine, Shandong Academy of Agricultural Sciences, Jinan 250100, China; hejunmin330@163.com (J.H.); weichenchen1989@126.com (C.W.); gpzhangsaas@163.com (G.Z.); m2797611850@163.com (J.M.); 2Key Laboratory of Livestock and Poultry Multi-Omics of MARA, Jinan 250100, China; 3College of Animal Science, Xinjiang Agricultural University, Urumqi 830052, China; au-huangxixia@163.com (X.H.); lixueli1126@163.com (X.L.); yangcunming0405@163.com (C.Y.); 15294192208@163.com (W.Z.)

**Keywords:** hair follicle development, sheep dermal fibroblasts, cotargeting, proliferation, migrate, apoptosis

## Abstract

Wool is produced and controlled by hair follicles (HFs). However, little is known about the mechanisms involved in HF development and regulation. Sheep dermal fibroblasts (SDFs) play a key role in the initial stage of HF development. Analyzing the molecular mechanism that regulates early HF development in superfine wool sheep is of great importance for better understanding the HF morphogenesis process and for the breeding of fine wool sheep. Here, we show that two microRNAs (miRNAs) affect the development of HFs by targeting two genes that are expressed by SDFs. Meanwhile, the overexpression and inhibition of oar-miR-23b and oar-miR-133 in SDFs cells and cell proliferation, apoptosis, and migration were further detected using a CCK-8 assay, an Annexin V-FITC assay, a Transwell assay, and flow cytometry. We found that oar-miR-23b, oar-miR-133, and their cotarget genes *TGFβ2* and *NOTCH1* were differentially expressed during the six stages of HF development in superfine wool sheep. Oar-miR-23b and oar-miR-133 inhibited the proliferation and migration of SDFs and promoted the apoptosis of SDFs through *TGFβ2* and *NOTCH1*. oar-miR-23b and oar-miR-133 inhibited the proliferation and migration of SDFs by jointly targeting *TGFβ2* and *NOTCH1*, thereby inhibiting the development of superfine wool HFs. Our research provides a molecular marker that can be used to guide the breeding of ultrafine wool sheep.

## 1. Introduction

Wool, which is a pure natural textile raw fiber material, has a high economic value and is increasingly favored by modern textile enterprises and consumers. The output of domestic fine wool, especially high-quality superfine wool, is far from meeting the processing needs of the textile market. Subo Merino is a kind of wool sheep that is independently bred in China, and its wool has a fineness of 17–19 μm. This is a new breed of sheep that produces worsted superfine wool. Improving the yield and quality of wool has always been a focus of research related to fine wool sheep breeding. Wool is produced by hair follicles (HFs), which are complex organs located in skin tissues. The first stage of HF development is the proliferation of epidermal cells to form a placode; under this structure, dermal cells accumulate, and the two cell types grow together downwards, towards the dermis, to form a dermal condensate [1]. Gradually, dermal cells enter the epithelial bud to form the anterior papilla, and, finally, as the HF lengthens and enters the dermis, epithelial glomerulus cells envelop the anterior papilla [1]. The first HFs that are formed are primary follicles, followed by secondary follicles and then the secondary-derived follicles which branch from the secondary follicles [2,3,4]. Sheep primary follicles develop mainly during the embryonic stage and complete their development process and reach maturity before birth; thus, no new primary follicles are produced at birth [5]. Studies have shown that primary follicles begin to develop on day 65 of the embryonic period (E65), secondary follicles begin to form on E85, secondary-derived follicles form on E105, and HFs are basically mature on E135 [2]. Mature HFs begin to cycle through the process of hair growth (anagen), catagen, and telogen [6,7]. HF morphogenesis is an extremely complex biological process that is regulated by many signaling pathways [8,9,10,11] (e.g., the TGF-β, Notch, Hippo, and Wnt signaling pathways). The addition of Wnt signaling in dermal papilla cells is considered a key factor in stimulating hair growth. Mesenchymal stem cell-derived signaling and the growth factors obtained by platelets influence hair growth through cellular proliferation to prolong the anagen phase (FGF-7), induce cell growth (ERK activation), stimulate hair follicle development (β-catenin), and suppress apoptotic cues (Bcl-2 release and Akt activation) [9]. It has been shown that TGF-β2 signaling is necessary to transiently induce the transcription factor Snail and activate the Ras-mitogen-activated protein kinase (MAPK) pathway in the bud. Rendl et al. identified BMP signaling as a determinant of stem cell-activating hair-inducing cell fate [12]. It is also affected by epigenetics [13] (e.g., DNA/RNA methylation) and noncoding RNAs [2] (e.g., lncRNAs, circRNAs, miRNA), especially the miRNA.

miRNAs are noncoding RNAs that are 22 nucleotides (nt) long and play a role in negatively regulating the post-transcriptional modification of messenger ribonucleic acids (mRNAs) in various biological processes [14]. An increasing number of studies have confirmed that miRNA molecules are involved in a variety of biological processes, including growth and development, cell differentiation, apoptosis, lipid metabolism, hormone secretion, signal transduction, and stress responses [15]. The dysregulation of these processes can lead to hair loss or skin problems that affect a person’s appearance in mild cases and are life-threatening in severe cases. miRNAs are closely related to our lives and are worthy of further study. While much is known about the general biological functions of miRNAs, such as their interactions with the RNA-induced silencing complex, many important questions remain unanswered, especially those regarding their functions in HFs. miRNAs can regulate the expression of HF development-related genes, thereby changing the phenotype and fate of epithelial cells, fibroblasts, dermal papilla cells, and HF stem cells. HF development and morphological changes are mainly controlled by these cells [16,17,18,19]. Some studies have also shown that miR-21 [20], miR-31 [21], and miR-214 [22,23] play important roles in keratinocytes. It had also been found that miR-21 affects HF development in superfine wool sheep and small-tailed Han wool sheep [24]. Previous studies have shown that miR-218-5p passes the Wnt/β-catenin signaling pathway and targets *SFRP2* to regulate the development of skin and HFs [19]. In terms of research on the role of miRNAs in HF development, very in-depth research has been conducted on cashmere goats. CircRNA-1926 enhances the expression of *CDK19* by sponging miR-148a/b-3p and promotes the differentiation of secondary HF stem cells into HF lines in cashmere goats [25]. Lv et al. [26] revealed that miR-148a and miR-10a can inhibit the proliferation of Hu wool papilla cells and are related to the growth and development of HFs. miR-195-5P regulates the ability of dermal papilla cells to induce HF formation by inhibiting Wnt/β-catenin activation [27]. In the HF cycle of Liaoning cashmere goats, miR-let-7a was found to regulate the expression of the *C-myc*, *FGF5*, and *IGF1R* genes, thereby affecting the development of HFs in said goats [28]. Studies have also shown that miR-149 [29], miR-205 [30,31], miR-125b [18,32], miR-214 [22,23], and miR-218 [17] all affect the differentiation and development of HF stem cells. Interestingly, some studies have shown that miR-128 regulates the differentiation of HF mesenchymal stem cells into smooth muscle cells by targeting the major transcription regulator of TGF-β, namely, *Smad2* [17].

HFs are a source of mixed cell populations because they contain cells from the epidermis and the dermis. During embryonic development, a series of mutually coordinated signals between epithelial cells from the epidermis and mesenchymal cells from the dermis trigger the formation of HFs [33,34]. Dermal fibroblasts (DFs) are mesenchymal cells that are found between the skin epidermis and subcutaneous tissues [35,36,37]. The mesenchymal cells of the dermis play a vital role in the formation of DFs in fetal HFs, and they play an equally important role in regulating their cyclic growth, rest, and regression phases in adults [38,39]. Even within a single tissue, fibroblasts exhibit remarkable functional diversity. Fibroblasts engage in fibroblast–epidermal interactions during hair development and in interfollicular regions of the skin [40]. The fibroblasts of skin connective tissue are derived from two distinct lineages [40,41,42]. The developing dermis undergoes fate restriction: the cells in the upper dermis give rise to the dermal papilla, the arrector pili muscle, and papillary fibroblasts, while the cells in the lower dermis give rise to the reticular dermis and the subdermal and adipocyte layers. The upper dermis is required for HF formation [41,42]. Fibroblasts play a crucial role in cutaneous wound repair. This also explains why trauma is associated with the formation of extracellular matrix component-rich scar tissues that lack HFs [43,44,45]. Sustained activation of the canonical Wnt pathway in the adult epidermis induces the growth of existing HFs (anagen), the formation of new HFs [46], the proliferation of fibroblasts, and the remodeling of the dermal extracellular matrix [47,48]. Since epidermal Wnt signaling promotes HF growth, the expansion of the lower dermis may explain why the skin adipocyte layer increases in thickness during aging. Dermal remodeling in response to epidermal β-catenin activation results in the expansion of the upper and lower dermis, and the increase in the upper papillary dermis allows the formation of new HFs [41,42]. At present, there are few reports on the effects of miRNAs on sheep dermal fibroblasts (SDFs).

Wool is an important textile material that is produced by HFs. During the development of HFs in ultrafine wool sheep, the process of embryonic HF development determines the yield and quality of the wool that will be produced by the sheep in adulthood. For example, the diameter, curvature, and density of wool fibers are related to the size of the wool substrate and the number of secondary hair follicles, and these characteristics have a high economic value in the wool industry. The structuring, functioning, and morphogenesis of HFs are complex biological processes, and the mechanism by which HF morphogenesis affects the traits of the wool that is produced by superfine wool sheep needs to be further studied. Therefore, analyzing the molecular mechanism that regulates the early development of HFs in superfine wool sheep is important for better understanding the HF morphogenesis process and for breeding sheep for wool-specific traits. Based on our previous research, we selected miRNAs (oar-miR-23b and oar-miR-133) [49] that are related to the induction of HF differentiation and key genes (*TGFβ2* and *NOTCH1*) [50] that are related to HF development for further exploration in this study. We knocked out and overexpressed oar-miR-23b and oar-miR-133 in SDFs and used bioinformatics, Cell Counting Kit-8 (CCK-8, Beyotime, Shanghai, China), reverse transcription quantitative real-time PCR (RT‒qPCR), Western blot, and flow cytometry to determine the effects of miRNAs on cell proliferation, apoptosis, migration, and cycle progression. Similarly, we also tested and analyzed miRNAs that target *TGFβ2* and *NOTCH1*. We hoped to provide a molecular theoretical basis for understanding and revealing the genetic mechanism regulating fine wool follicle development.

## 2. Materials and Methods

### 2.1. Animal Selection and Skin Tissue Preparation

Subo Merino sheep is a subtype of the Merino breed of sheep in China that is known for its high survival rate and excellent quality and yield of wool. A sheep herd located in the Kechuang Animal Husbandry Breeding Center, Xinjiang, China, was selected for testing. From this flock, twenty healthy ewes were artificially inseminated with fresh sperm from the same ram. Insemination occurred on embryonic day 0 (E0). Previous reports have described the methods used for embryonic skin tissue collection on E65, E85, E105, and E135 [49,50]. The method for the collection of skin tissues on postnatal days 7 and 30 (D7 and D30) has been described previously [49,50]. For each of the six developmental stages represented by the groups, three biological replicates were generated. All eighteen skin tissue samples were stored at −80 °C. Our previous article describes the details of the DE-miRNA sequence analysis [49].

### 2.2. Cell Culture and Transfection

Primary sheep dermal fibroblasts were cultured as in a previous study [49]. SDFs were prepared from one newborn lamb on postnatal day 7 (local anesthesia before surgery). Surface hair was removed with a blade, and the skin was wiped with cotton balls soaked in 75% alcohol. Then, Lidocaine aerosol (Lishuka, A1004157, Shanghai Sine Pharmaceutical Laboratories Co., Ltd., Shanghai, China) was used to spray the exposed skin surface, and, after waiting for 2 min, 2 cm^2^ of skin tissue was collected with a scalpel and scissors. Then, 100 IU/mL penicillin and 0.1 mg/mL streptomycin (2:100) were added to phosphate-buffered saline (PBS) at 4 °C. Briefly, the skin tissues were incubated in 0.25% trypsin at 4 °C overnight. The tissue blocks were cut into 0.8 mm^2^ tissue blocks with ophthalmic scissors, broken up with a pipette, and placed in a Petri dish that contained sterile medium (Dulbecco’s modified Eagle’s medium, DMEM) (Invitrogen, Carlsbad, CA, USA), basal medium supplemented with double antibodies, and 10% fetal bovine serum (FBS) (Invitrogen, Carlsbad, CA, USA). The cells were incubated at 37 °C with 5% CO_2_, and then the medium was changed every three days to observe cell growth in the culture dish. If fibroblasts were freed, the tissue blocks could be washed away. Subsequently, adherent fibroblasts were reseeded in new culture plates with EMEM supplemented with 10% FBS, 100 IU/mL penicillin, and 0.1 mg/mL streptomycin. The cells were passaged when cell confluence reached a value of more than 80%.

HEK-293T cells were provided by the Key Laboratory of Livestock and Poultry Multi-omics of MARA (Ministry of Agriculture and Rural Affairs). HEK-293T cells were cultured at 37 °C in DMEM (Invitrogen, Carlsbad, CA, USA), supplemented with 10% FBS (Invitrogen, Carlsbad, CA, USA), 1.5 mM l-glutamine (Invitrogen, Carlsbad, CA, USA), 100 U/mL penicillin (Invitrogen, Carlsbad, CA, USA), and 100 mg/mL streptomycin (Invitrogen, Carlsbad, CA, USA), in a humidified incubator in an atmosphere containing 5% CO_2_ (Thermo, Waltham, MA, USA). None of the cell lines used in this paper were listed in the database of commonly misidentified cell lines that is maintained by the ICLAC. All the cell lines were free of mycoplasma contamination. Adherent cells were passaged daily with 0.05% trypsin-EDTA (Invitrogen, Carlsbad, CA, USA).

### 2.3. RT‒qPCR

Total RNA was extracted using TRIzol reagent (Invitrogen, Carlsbad, CA, USA). Poly-A tails were added to the miRNAs according to the protocol of the Poly (A) Tailing Kit (Ambion, Austin, TX, USA). The PrimeScript^TM^ RT Reagent Kit with gDNA Eraser (Takara, Kusatsu, Japan) and gene-specific primers or random primers were used to generate cDNA. RT‒qPCR was performed with a CFX96^TM^ Real-Time System (Bio-Rad, Hercules, CA, USA) using SYBR^®^ Green (Takara, Kusatsu, Japan) and the miRcute Plus miRNA qPCR Kit (SYBR Green) (TIANGEN, Beijing, China). Glyceraldehyde phosphate dehydrogenase (*GAPDH*) and U6 snRNA were used as endogenous controls for mRNA and miRNA, respectively. The thermal cycling conditions used for mRNA RT‒qPCR were 95 °C for 30 s, followed by 40 cycles of 95 °C for 5 s and 60 °C for 30 s. The thermal cycling conditions used for miRNA RT‒qPCR were 95 °C for 15 min, followed by 45 cycles of 94 °C for 20 s and 60 °C for 34 s. The specificity of the SYBR green PCR signal was confirmed by a melting curve analysis. There were three biological and technical replicates. The comparative Ct method was used to calculate the relative expression of the target RNAs. The primers were synthesized by Shanghai Sangong Biology Co., Ltd. (Shanghai, China). The primer sequences are displayed in Appendix A. Sequence 2^−ΔΔCt^ indicated the change in the expression of the gene of interest between the experimental group and the control group.

### 2.4. Western Blotting Analysis

The total protein was extracted from the SDFs using a radioimmunoprecipitation assay lysis buffer (Beyotime, Shanghai, China). Protein concentration was determined using a bicinchoninic acid (BCA) kit (Thermo, Waltham, MA, USA). The proteins were separated by polyacrylamide gel electrophoresis. Next, the separated proteins were transferred onto polyvinylidene fluoride membranes (Millipore, Boston, MA, USA), which were blocked in 5% BSA at room temperature for 1 h. The membranes were incubated overnight at 4 °C with the following primary antibodies (Abclonal, Wuhan, Hubei, China): anti-Notch1 rabbit mAb (A19090, 1:10,000), anti-TGFβ2 rabbit pAb (A3640, 1:10,000), anti-Cyclin D1 rabbit pAb (A11022, 1:10,000), anti-Bax rabbit pAb (A15646, 1:10,000), anti-p53 rabbit pAb (A11232, 1:10,000), anti-CDKN1A/p21CIP1 rabbit pAb (A1483, 1:10,000), and anti-GAPDH (A19056, 1:10,000). After three washes, the membranes were incubated with diluted horseradish peroxidase (HRP)-labeled goat anti-rabbit immunoglobulin G (IgG) (AS014, 1:10,000, Abclonal, Wuhan, Hubei, China) for 1 h at room temperature. Then, the membranes were visualized with enhanced chemiluminescence substrates (BL520A, Biosharp, Beijing, China). The ImageJ 1.48u software (Bethesda Softworks, LLC, Rockville, MD, USA) was used for the protein quantification analysis, and GAPDH was used as the internal reference.

### 2.5. Prediction and Enrichment Analysis of Target Genes

miR-23b and miR-133 target genes were predicted through the online websites TargetScan (http://www.targetscan.org, accessed on 4 March 2023), RNAhybrid (https://bibiserv.cebitec.uni-bielefeld.de/rnahybrid/%EF%BC%89, accessed on 4 March 2023), PicTar (https://pictar.mdc-berlin.de, accessed on 4 March 2023), starBase (http://starbase.sysu.edu.cn/, accessed on 4 March 2023), and RNA22.

GO and KEGG functional analyses of target genes were performed using Database for Annotation, Visualization, and Integrated Discovery (DAVID) (https://david.ncifcrf.gov/, accessed on 18 April 2023). GO is a database established by the Gene Ontology Consortium that is used for the functional annotation and classification of target genes. A KEGG functional analysis was used to perform the functional annotation and classification of pathways in the KEGG database for the target genes. The method of the KEGG pathway’s functional enrichment analysis was similar to that of the GO functional enrichment analysis. Based on the false discovery rate (FDR), we determined the GO terms and metabolic pathways that were significantly associated with the gene lists. An FDR < 0.05 was applied to significant target genes associated with GO terms or KEGG pathways.

### 2.6. Luciferase Reporter Assay

Total RNA was extracted from sheep skin according to the instructions of the TRIzol kit (Invitrogen, USA). The extracted RNA was then reverse transcribed into complementary deoxyribose nucleic acid (cDNA) according to the instructions of the One StepPrimeScript^®^ miRNA cDNA Synthesis Kit (TaKaRa, Kusatsu, Japan). The 3′UTR fragments of *NOTCH1* and *TGFβ2* that contained putative oar-miR-23b/oar-miR-133 binding sites were amplified from sheep genomic DNA using forward and reverse primers containing XhoI and NotI restriction sequences, respectively. The amplified fragments were cloned at the XhoI and NotI sites downstream of the CV40 promoter-driven Renilla luciferase cassette in psiCHECK2 (Promega, Madison, WI, USA). The dual-luciferase reporter vector carrying the wild-type (WT) 3′UTR of *NOTCH1*/*TGFβ2* and the 3′UTR sequences with mutations (MUTs) in the miR-23b and miR-133 binding sites were separately constructed. The primers and sequences used for the construction of wild-type and mutant vectors are shown in Appendix A. HEK-293T cells with good growth were selected and seeded in 24-well plates at a density of 1 × 10^4^ cells/well, followed by a culture in an incubator at 37 °C with 5% CO_2_ and saturated humidity. When the cells grew to approximately a 50% confluence, cell transfection was performed according to the instructions of the Lipofectamine 3000 transfection kit (Invitrogen, USA). miRNA-23b-mimic (pro-miR-23b), psiCHECK2-*NOTCH1*-WT1 or psiCHECK2-*NOTCH1*-MUT1, psiCHECK2-*TGFβ2*-WT1 or psiCHECK2-*TGFβ2*-MUT1, and miRNA-NC (negative control) were transfected into HEK-293T cells. Similarly, miR-133 mimic (pro-miR-133), psiCHECK2-*NOTCH1*-WT2 or psiCHECK2-*NOTCH1*-MUT2, psiCHECK2-*TGFβ2*-WT2 or psiCHECK2-*TGFβ2*-MUT2, and miRNA-NC (negative control) were transfected into HEK-293T cells. The medium was replaced with fresh medium 6 h after transfection. Twenty-four hours after transfection, the relative luciferase activities were determined using the Dual-Glo Luciferase Assay System (Promega, USA). The assay was performed in triplicate for three independent trials.

### 2.7. Cell Transfection

SDFs were cultured at 37 °C in DMEM (Invitrogen, Carlsbad, CA, USA) supplemented with 10% FBS (Invitrogen, Carlsbad, CA, USA), 1.5 mM L-glutamine (Invitrogen, Carlsbad, CA, USA), 100 U/mL penicillin (Invitrogen, Carlsbad, CA, USA), and 100 mg/mL streptomycin (Invitrogen, Carlsbad, CA, USA) in a humidified incubator in an atmosphere containing 5% CO_2_ (Thermo, Waltham, MA, USA). When the cells grew to approximately a 50% confluence, they were transfected with 200 nM synthetic oar-miR-23b inhibitor (anti-miR-23b) and oar-miR-133 inhibitor (anti-miR-133), oar-miR-23b mimic (pro-miR-23b) and oar-miR-133 mimic (pro-miR-133), or miRNA negative controls (mimic-NC and inhibitor-NC), using Lipofectamine 3000 (Invitrogen, Carlsbad, CA, USA). The cells were harvested 48 h after transfection and used for further analyses.

### 2.8. Cell Proliferation Assay

A CCK-8 assay (Beyotime, Shanghai, China) was used to analyze the cell proliferation rate. Oar-miR-23b mimic, oar-miR-23b inhibitor, mimic-NC, inhibitor-NC, oar-miR-133-mimic, and oar-miR-133 inhibitor were transfected. Twenty-four hours after the SDFs were transfected, they were digested with EDTA-0.25% trypsin, and 600 µL/well (approximately 2 × 10^3^ cells/mL) was seeded in a 24-well plate for culture. Four time points were established for each group: 24 h, 48 h, 72 h, and 96 h. After being cultured for 24, 48, 72 h, and 96 h, 60 µL of CCK-8 solution was administered to each well. Then, after being cultured for another 2 h, the supernatants were transferred to a 96-well plate (100 µL/well), and six replicates were established for each sample. Then, the absorbance was measured at 450 nm with a microplate reader, and the cell proliferation curve was generated according to the OD values.

### 2.9. Cell Apoptosis Analysis by Annexin V-FITC Staining

SDFs in the logarithmic phase of growth and in good growth conditions were collected, seeded in six-well plates at a concentration of 5 × 10^5^ cells/well, and cultured overnight at 37 °C in a 5% CO_2_ incubator. The cells were treated according to the following groups: oar-miR-23b mimic; oar-miR-23b inhibitor; mimic-NC; inhibitor-NC; oar-miR-133-mimic; and oar-miR-133 inhibitor. After 48 h of treatment, the instructions of the Annexin V-FITC/PI apoptosis detection kit were followed, and flow cytometry was used for the analysis.

### 2.10. Transwell Cell Migration Assay

SDFs were harvested 48 h post transfection (oar-miR-23b mimic, oar-miR-23b inhibitor, mimic-NC, inhibitor-NC, oar-miR-133 mimic, or oar-miR-133 inhibitor), and the cells were counted. Transwell chambers (Corning, NY, USA) were then placed in 24-well plates. A total of 200 μL of cell suspension (3 × 10^5^ cells/mL) was added to the Transwell chamber. The cells were incubated for 24 h in a humidified incubator (37 °C/5% CO_2_). The SDFs were then fixed with cold 70% ethanol and incubated at room temperature for 1 h. Then, the cells were stained with 0.5% crystal violet dye, photographed, and counted (10 × 10, 3 pictures/group).

### 2.11. Flow Cytometry Analysis

SDFs were harvested 48 h post transfection (oar-miR-23b mimic, oar-miR-23b inhibitor, mimic-NC, inhibitor-NC, oar-miR-133 mimic, or oar-miR-133 inhibitor). The cells were subsequently resuspended in precooled PBS and fixed overnight at 4 °C in precooled 70% ethanol. The SDFs were washed three times with PBS, and, then, the cells were centrifuged for 5 min at 1500 rpm and resuspended in 200 μL of PBS. To each sample, 10 μL of RNase (10 mg/mL) was added and then incubated at 37 °C for 30 min. Then, 10 μL of RNaseA (400 μg/mL) was added to these cells and incubated for 30 min at 4 °C in the dark. Finally, cell cycle progression was analyzed by flow cytometry.

### 2.12. Statistical Analysis

Statistical analyses were performed using the GraphPad Prism 7 software (GraphPad Software, San Diego, CA, USA). The experimental results are expressed as the mean ± standard deviation (mean ± SD). A *t* test was used to analyze the differences between the two groups. One-way ANOVA was performed to analyze the differences among different groups, followed by a post hoc test (least significant difference). Significant differences were denoted by * *p* < 0.05, and extremely significant differences were indicated by ** *p* < 0.01. The plots were generated using the GraphPad Prism software (version 7).

## 3. Results

### 3.1. DE-miRNA Clustering and RT‒qPCR Analysis

In our previous research, 87 differentially expressed miRNAs (DE-miRNAs) were identified by sequencing analysis [49]. We found that five DE-miRNAs (miR-23b, miR-133, miR-381-5p, miR-381-3p, and miR-655-3p) were differentially expressed in the first three stages of growth (E65, E85, and E105) (Figure 1A). Combined with previous research results of the mRNA–miRNA network [49], we selected miR-23b and miR-133 for further study. To elucidate the roles of these two miRNAs in hair follicle development, we performed RT‒qPCR analysis on the skin tissues that had been isolated during the six stages of hair follicle development (E65, E85, E105, E135, D7, and D30). The results showed that miR-23b, miR-133, *TGFβ2*, and *NOTCH1* were differentially expressed in the six phases. The expression of miR-23b showed an upward trend (Figure 1B); the expression trend of miR-133 first increased and then decreased, and its highest expression peak occurred on E105 (Figure 1C). *TGFβ2* expression peaked on E85 and reached its lowest point on postnatal days 7 and 30 (Figure 1D). *NOTCH1* was expressed at its lowest level on E65 (Figure 1E).

### 3.2. Functional Enrichment Analysis of Oar-miR-23b and Oar-miR-133 Target Genes

To further analyze the functions of oar-miR-23b and oar-miR-133, in this study, a gene ontology (GO) enrichment analysis was performed on their target genes. The top 10 GO terms are shown. According to the GO analysis, we found that, in their cellular composition (CC), oar-miR-23b’s and oar-miR-133’s target genes were significantly enriched in the cytosol, the nucleus, the nucleoplasm, the Golgi apparatus, the cytoplasm, and glutamatergic synapses (Appendix A, *p* < 0.01). However, only oar-miR-23b’s target genes were enriched in the neuronal cell body, dendrites, early endosomes, and the perinuclear region of the cytoplasm (Appendix A). Similarly, only the target genes of oar-miR-133 were enriched in the basolateral plasma, the membrane, the endoplasmic reticulum membrane, the cell surface, and the ubiquitin ligase complex (Appendix A). In terms of molecular functions (MFs), the target genes of both miRNAs were significantly enriched in transcriptional activator activity, RNA polymerase II transcription regulatory region sequence-specific binding, ATP binding, RNA polymerase II core promoter proximal region sequence-specific DNA binding, transcription factor activity, sequence-specific DNA binding, and metal ion binding (Appendix A, *p* < 0.01). Only the target genes of oar-miR-23b were enriched in DNA binding, sequence-specific DNA binding, RNA polymerase II transcription factor activity, sequence-specific DNA binding, mRNA 3′-untranslated region (3′UTR) binding, and SMAD binding (Appendix A). Only the target genes of oar-miR-133 were enriched in magnesium ion transmembrane transporter activity, GTPase activator activity, protein kinase binding, protein homodimerization activity, and small GTPase binding (Appendix A). We further analyzed the biological processes (BPs) and found that the target genes of oar-miR-23b and oar-miR-133 were enriched in a negative regulation of transcription from the RNA, the polymerase II promoter, the cellular response to the leukemia inhibitory factor, peptidyl-threonine phosphorylation, and peptidyl-serine phosphorylation (Appendix A, *p* < 0.01). The oar-miR-23b target genes were individually enriched in endocytic recycling, microtubule cytoskeleton organization, transcription, DNA templating, protein import into the nucleus, protein autophosphorylation, and positive regulation of gene expression (Appendix A). The target genes of oar-miR-133 were also enriched in the transforming growth factor beta receptor signaling pathway, the Wnt signaling pathway, the positive regulation of cytoplasmic mRNA processing body assembly, pathway-restricted SMAD protein phosphorylation, endocytosis, and intracellular signal transduction in these biological processes (Appendix A). In conclusion, the target genes of oar-miR-23b and oar-miR-133 shared 50% of their enriched GO terms, indicating that they are not only unique but also similar.

According to the Kyoto Encyclopedia of Genes and Genomes (KEGG) analysis, we found that the oar-miR-23b and oar-miR-133 target genes were significantly enriched in 14 pathways (Appendix A, *p* < 0.01). These pathways included axon guidance, the MAPK signaling pathway, pathways in cancer, including breast and pancreatic cancer and proteoglycans, the Ras signaling pathway, the FoxO signaling pathway, the signaling pathways regulating the pluripotency of stem cells, endocytosis, the Wnt signaling pathway, the PI3K-Akt signaling pathway, the ephingolipid signaling pathway, and the phosphatidylinositol signaling system. Additionally, 16 oar-miR-23b target genes were individually enriched in pathways including focal adhesion, transcriptional misregulation in cancer, the ErbB signaling pathway, the mTOR signaling pathway, renal cell carcinoma, chronic myeloid leukemia, non-small-cell lung cancer, EGFR tyrosine kinase inhibitor resistance, microRNAs in cancer, etc. (Appendix A). Only the oar-miR-133 target genes were enriched in the regulation of the actin cytoskeleton, the oxytocin signaling pathway, the cGMP-PKG signaling pathway, the Rap1 signaling pathway, colorectal cancer, platelet activation, cholinergic synapse, and the calcium signaling pathway (Appendix A). Interestingly, the target genes of both miRNAs were significantly enriched in the MAPK, Wnt, and PI3K-Akt signaling pathways.

### 3.3. Oar-miR-23b and Oar-miR-133 Cotarget NOTCH1/TGFβ2

To assess the interaction between oar-miR-23b and oar-miR-133 and their target genes, we constructed the relevant vectors. The results of DNA sequencing indicated that the construction of the vector was successful (Figure 2A). The interaction between each target gene and oar-miR-23b and oar-miR-133 was determined with a dual-luciferase reporter validation system, and the results showed that oar-miR-23b and oar-miR-133 inhibited the expression of their target genes. There was no significant difference in the expression of *NOTCH1* and *TGFβ2* between the group that had been cotransfected with the mimics and the mutant sequence (MUT group) and the group that had been cotransfected with the mimics and the control sequence. However, the fluorescence signal of the group that had been cotransfected with the mimics and the wild-type sequence (WT group) was significantly lower (Figure 2B,C, *p* < 0.01).

### 3.4. Oar-miR-23 and Oar-miR-133 Inhibit NOTCH1/TGFβ2 Expression

To investigate the overexpression and inhibition of oar-miR-23b and oar-miR-133 in SDF cells, we utilized RT-qPCR and WB at the mRNA level and protein level. The mRNA expression levels of *NOTCH1* and *TGFβ2* after the transfection of oar-miR-23b and oar-miR-133 were measured by RT‒qPCR. The results showed that the expression levels of *NOTCH1* and *TGFβ2* after oar-miR-23b overexpression were opposite to those of oar-miR-23b, and the overexpression of oar-miR-133 resulted in opposite expression patterns to those of oar-miR-133 (Figure 3A,B, *p* < 0.01). More importantly, the expression of *NOTCH1* and *TGFβ2* in the oar-miR-23b overexpression group was significantly downregulated compared with that in the negative control group (*p* < 0.01). A significant downward trend was observed after the overexpression of oar-miR-133 compared to the negative control (*p* < 0.01). A significant difference was observed between the inhibition group and the control group (*p* < 0.01). The inhibition group exhibited significantly higher expression of *NOTCH1* and *TGFβ2* than the control group (*p* < 0.01). *NOTCH1* and *TGFβ2* have molecular weights of 125 and 48 kDa, respectively (Figure 3C), and the expression levels of *NOTCH1* and *TGFβ2* in the oar-miR-23b overexpression group significantly decreased. The expression levels of both *NOTCH1* and *TGFβ2* were significantly reduced in the oar-miR-133 overexpression group. However, the inhibition of oar-miR-23b and oar-miR-133 significantly increased the protein expression levels of these proteins (Figure 3D,E, *p* < 0.01).

### 3.5. Effects of Oar-miR-23b and Oar-miR-133 on Pathway-Related Gene Expression

To further understand the regulatory roles of the oar-miR-23b and oar-miR-133 genes, we selected star genes from the PI3K-Akt, MAPK, TGF-β, Hippo, and WNT pathways for RT‒qPCR analysis. We found that the overexpression of oar-miR-23b significantly inhibited the expression levels of DKK1, FZD3, FZD6, HOXC13, LGR4, TGFβ1, and WNT5A (*p* < 0.01). The inhibition of oar-miR-23b significantly inhibited the expression of the BMP2, FZD3, FGF7, and INHBA genes (*p* < 0.01) but significantly upregulated the expression of the SAMD5 and WNT5A genes (*p* < 0.01). However, both oar-miR-23b overexpression and inhibition reduced the expression of BMP2 and FZD3, and the downregulation caused by oar-miR-23b overexpression was more pronounced (Figure 4A). We also found that the overexpression of oar-miR-133 significantly reduced the expression levels of DKK1, FZD3, FZD6, TGFβ1, TGFβ3, and WNT5A (*p* < 0.01). The knockdown of oar-miR-133 significantly reduced the expression levels of FGF7 and BMP2 and promoted the expression of TGFβ1, TGFβ3, and WNT5A (Figure 4B, *p* < 0.01).

### 3.6. Oar-miR-23b and Oar-miR-133 Enhance Apoptosis and Inhibit Proliferation of SDFs

To investigate the effects of the overexpression/inhibition of miR-23b and miR-133 on cell proliferation and apoptosis in SDFs, a CCK-8 cell proliferation assay was performed and apoptosis was analyzed by flow cytometry. The mRNA expression levels of the transfected miR-23b and miR-133 were measured by RT‒qPCR. The results showed that miR-23b and miR-133 expression in the overexpression (mimic) group was significantly higher (Figure 5A,B, *p* < 0.01). The transfected cells were used in subsequent experiments. The proliferation of SDFs that had been transfected with oar-miR-23b mimic/mimic-NC and oar-miR-23b inhibitor/inhibitor-NC was assessed by the CCK-8 method. Compared with the negative control group (mimic-NC), the proliferation of cells that had been transfected with the oar-miR-23b mimic was significantly decreased (*p* < 0.01). However, the proliferation of inhibitor-treated cells that had been transfected with oar-miR-23b was significantly higher than that of inhibitor-NC-treated cells (*p* < 0.05). The results showed that oar-miR-23b inhibited the proliferation of SDFs (Figure 5C). The proliferation of cells that had been transfected with the oar-miR-133 mimic was significantly lower than that of the cells that had been transfected with the mimic-NC (*p* < 0.01). However, the proliferation of inhibitor-treated cells that had been transfected with oar-miR-133 was significantly higher than that of inhibitor-NC-treated cells (*p* < 0.05). The results showed that oar-miR-133 inhibited the proliferation of SDFs (Figure 5D). The apoptosis of oar-miR-23b-transfected SDFs was analyzed by flow cytometry. Compared with the control group, the treatment group showed an increased cellular apoptosis rate. Compared with mimic-NC, overexpression of endogenous oar-miR-23b increased the apoptosis rate of SDFs. The inhibition of endogenous oar-miR-23b reduced the apoptosis rate of SDFs compared with inhibitor-NC (Figure 5E). A statistical analysis showed that the overexpression of oar-miR-23b significantly (*p* < 0.01) promoted the apoptosis of SDFs, while the inhibition of oar-miR-23b expression significantly (*p* < 0.01) inhibited the apoptosis of SDFs (Figure 5F). The treatment group exhibited increased apoptosis relative to the control group. Oar-miR-133 overexpression increased the number of SDFs undergoing apoptosis. Figure 5G shows that the inhibition of oar-miR-133 could reduce SDF apoptosis. The overexpression of oar-miR-133 significantly (*p* < 0.01) promoted the apoptosis of SDFs, while the inhibition of oar-miR-133 expression significantly (*p* < 0.01) inhibited the apoptosis of SDFs (Figure 5H).

To investigate the effects of the overexpression/inhibition of miR-23b and miR-133 on the cell cycle of SDFs, cell cycle-related protein expression was measured by Western blotting. Cyclin D1 can promote cell proliferation, CDKN1A can inhibit cell proliferation, and BAX and P53 can promote cell apoptosis. To further understand the effect of oar-miR-23b on SDFs, we performed a Western blotting analysis of cell cycle-related protein expression. The results showed that the overexpression of oar-miR-23b effectively inhibited the protein expression of Cyclin D1 but significantly promoted the protein expression of CDKN1A, BAX, and P53. Conversely, the inhibition of oar-miR-23b effectively promoted the protein expression of Cyclin D1, while the inhibition of oar-miR-23b significantly promoted the protein expression of Cyclin D1 and inhibited the protein expression of CDKN1A, BAX, and P53 (Figure 6A–E). Combining the proliferation and apoptosis results, we can conclude that oar-miR-23b inhibits cell proliferation and promotes cell apoptosis. The apoptosis of SDFs can be promoted by oar-miR-133. The overexpression of oar-miR-133 effectively inhibited Cyclin D1 expression but significantly promoted CDKN1A, BAX, and P53 expression. In contrast, inhibiting oar-miR-133 effectively promoted Cyclin D1 protein expression (Figure 6A,C). Cyclin D1 expression was significantly increased by the inhibition of oar-miR-133, whereas CDKN1A, BAX, and P53 protein expression was significantly suppressed by the inhibition of oar-miR-133 (Figure 6A,B,D,E). Based on our findings, we can conclude that oar-miR-133 inhibits cell proliferation and promotes apoptosis.

### 3.7. Oar-miR-23b and Oar-miR-133 Inhibit Cell Migration and Alter Cell Cycle Progression of SDFs

We examined the effects of altering oar-miR-23b and oar-miR-133 expression on the migration and cell cycle progression of SDFs. Our results show that the overexpression of oar-miR-23b and oar-miR-133 inhibited cell migration, while the inhibition of oar-miR-23b and oar-miR-133 promoted cell migration (Figure 7A–D). Both oar-miR-23b and oar-miR-133 inhibited SDF migration. Similarly, we also found that the overexpression of oar-miR-23b and oar-miR-133 significantly promoted the G1 and G2 phases of the cell cycle but significantly inhibited the S phase of the cell cycle. However, the results of inhibiting oar-miR-23b and oar-miR-133 were opposite. We concluded that the overexpression of oar-miR-23b and oar-miR-133 promoted cell cycle progression and increased the ratio of SDF cells in the G1/S phase (Figure 7E–H). In contrast, the inhibition of oar-mir-23b and oar-miR-133 resulted in a decrease in the number of cells in the G1/S phase. These findings indicate that the downregulation of oar-miR-23b and oar-miR-133 accelerates cell cycle progression in SDFs.

## 4. Discussion

During HF morphogenesis, intersecting signaling networks in epithelial and mesenchymal cells control the transcription, adhesion, polarity, and motility programs in these selected cell types. The dynamic changes in the nucleus and cytoplasm that occur during this period constitute the cornerstone of organ morphogenesis. There are two main challenges to understanding the mechanisms underlying specific processes in HFs: one challenge involves sequencing the time series of the external signals involved, and the other challenge involves analyzing how developing cells that are related to HF development translate these signals into downstream cellular remodeling, proliferation, and differentiation events. Our study provides some insights into how these events are coordinated during HF formation in developing skin. Combined with our previous research results, we found that oar-miR-23b and oar-miR-133 play key roles in the induction of HF differentiation (E65~E105), and they are expressed in skin tissue during the six stages of HF development [49]. We explored the roles of oar-miR-23b and oar-miR-133 and their target genes *NOTCH1* and *TGFβ2* in the postnatal development of SDFs. oar-miR-23b expression gradually increased in HF development and after birth and continued to be highly expressed 30 days after birth. These results show that oar-miR-23b plays a key regulatory role in the early differentiation of HFs until the maturation of HFs. Oar-miR-133 was upregulated from E65 to E105, with the highest expression on E105. This indicates that oar-miR-133 plays a key regulatory role in SD differentiation. The target genes of these two miRNAs, namely, *NOTCH1* and *TGFβ2*, also play roles in HF development, especially in the HF-induced differentiation stage (E65–E105).

In this study, we found that the overexpression of oar-miR-23b and oar-miR-133 significantly inhibited *TGFβ2* and *NOTCH1* expression in SDFs and inhibited the expression of genes related to the WNT, Hippo, and TGF-β signaling pathways. The overexpression of oar-miR-23b significantly inhibited the proliferation and migration of SDFs and promoted the apoptosis of SDFs. In the current literature, research on miR-23b and miR-133 has mainly focused on human diseases [51,52,53], while the impact of miR-23b and miR-133 on HF development and SDFs has not been reported. The coculture of DFs and keratinocytes modifies the activities of both cell types [54]. Keratinocytes induce the expression of *TGFβ2* by DFs. DFs regulate the production of laminins and type VII collagen by keratinocytes, possibly through TGF-β signaling [54]. Hu et al. [55] found that the expression of miR-23b in human immortalized keratinocytes is positively correlated with the concentration and time of *TGFβ1* exposure, and they proved that miR-23b accelerates the migration of human immortalized keratinocytes by downregulating the expression of *TIMP3*. Some studies have also shown that miR-23b-3p overexpression can enhance the expression of collagen type I, *COL1A1*, *COL3A1*, and *ACTA2* in atrial fibroblasts but has no significant effect on the proliferation and migration of atrial fibroblasts [56]. Skin fibrosis is a chronic debilitating feature of several skin diseases that leads to characteristic increases in dermal fibroblast proliferation and collagen deposition through the upregulation of components of the TGF-β/SMAD pathway. Andrew M et al. [57] found that miRNAs, including miR-29, miR-196a, and let-7a, as well as decreasing the transcription of miR-21, miR-23b, and miR-31, are involved in skin fibrosis. Chen et al. [58] found that, in bovine mammary epithelial cells and mouse mammary cells, miR-133a, which can specifically target *TGFβ2*, is most significantly downregulated in cadmium-treated bovine mammary epithelial cells, and circ08409 can regulate the proliferation, apoptosis, and inflammation of bovine mammary epithelial cells by binding to miR-133a, thereby reducing the inhibitory effect of miR-133a on *TGFβ2* expression. Other studies have shown that miR-133 affects rat cardiomyocyte apoptosis through *NOTCH1* [59]. Similarly, some studies have shown that miR-133 inhibits renal damage in diabetic nephropathy through the MAPK/ERK signaling pathway [60]. Xu et al. [61] found that the overexpression of miR-133 inhibits the growth of glioblastoma multiforme cells and increases the cellular apoptosis rate, while the knockdown of miR-133 increases the growth of cells and decreases the cellular apoptosis rate. The overexpression of miR-133 also inhibits the proliferation and migration of primary endothelial cells by targeting *FGFR1* [62]. These results are consistent with our study, which showed that the overexpression of oar-miR-23b and oar-miR-133 can inhibit the proliferation and migration of SDFs and promote cell apoptosis. Hair follicle development is also regulated by dermal hair papilla cells, and, in future studies, we will explore the mechanism of action of miR-23b and miR-133 further in dermal papilla cells.

*TGFβ2* was upregulated from E65 to E85, indicating its critical role in SF development. *TGFβ2* expression continuously decreased, beginning on E105. TGF-β2 acts as a coregulator between cells and performs multiple functions, including mediating the epithelial–mesenchymal transition, regulating cell proliferation, mediating endothelial fibrosis, and affecting cell apoptosis [63,64]. Sonic hedgehog and *TGFβ2* signaling also play important roles in hair follicle morphogenesis, but, in the absence of Lef-1, sonic hedgehog, or *TGFβ2*, compared to β-catenin-deficient skin [65,66], some hairy buds will still form. TGF-β is known to promote keratinocyte exit from the cell cycle [67]. However, unlike TGFβ1-deficient skin, which exhibits an extended phase of postnatal HF growth, *TGFβ2*-deficient skin exhibits embryonal impairment of follicle germination [65], but approximately 50% of *TGFβ2*-deficient shoots do not appear to be able to progress to the downward growth stage. This feature cannot be easily explained by the previously identified effects of TGF-β. miR-370-3p inhibits the proliferation and promotes the migration of epithelial cells and fibroblasts. However, it does not affect cell apoptosis. miR-370-3p inhibits the proliferation of epithelial cells and fibroblasts by targeting *TGFβR2* and *FGFR2*, thereby improving cell migration and ultimately regulating the fate of epithelial cells and DFs to form lamellar and dermal condensates, promoting HF morphogenesis [68]. *TGFβ2*, which is an intercellular coregulator, performs multiple functions, including mediating epithelial–mesenchymal transformation, regulating cell proliferation, mediating endothelial fibrosis, and affecting cell apoptosis [63,64]. Some studies have shown that chi-miR-199a-5p can inhibit the expression of *TGFβ2* in fibroblasts, and the *TGFβ2* gene is the target gene of chi-miR-199a-5p [69]. This is consistent with our results, according to which *TGFβ2* is targeted by oar-miR-23b and oar-miR-133, which can inhibit the expression of *TGFβ2* in SDFs.

*NOTCH1* is upregulated during HF differentiation, indicating that it is indispensable for HF development. Although the embryonic development of HFs can be achieved without NOTCH, their postnatal development requires complete Notch signaling in the hair bulb and outer root sheath (ORS) [70]. In HFs, NOTCH plays two roles: NOTCH controls the switching of cell fate of HF stem cells or their progenitor cells. In the hair bulb, NOTCH controls cell differentiation to ensure the normal development of each layer of the hair stem and inner root sheath (IRS). In hairballs, *NOTCH1* is expressed at both the mRNA [71] and protein [72] levels. However, it is not expressed in presumed ORS cells. In HFs, NOTCH functions via both cellular autonomous and cellular nonautonomous mechanisms, and it is involved in intercellular communication between adjacent layers. During embryonic development, *NOTCH1* mRNA expression is observed in the epidermis of the Mx, mainly in the invaginated inner cells of the epidermis, but not in the mesenchymal part [71,73]. During the growth period of mature HFs, *NOTCH1* is expressed in the inner cells of HFs and the basal cells of ORS [71,74]. The *NICD* (*NOTCH1* intracellular domain) that is produced after *NOTCH1* activation is strongly expressed in undifferentiated hair stromal cells and the cortex and keratinocytes of the hair stem, and a small number of cells are also expressed in the keratinocytes of the ORS and IRS [75,76]. In growing HFs, *NOTCH1* is expressed in the cuticle and dermal papilla of the IRS of the ORS [72]. In this study, we found that the *NOTCH1* gene was expressed in the skin of sheep during the six stages of HF development, and the highest expression was observed on E105, indicating that the gene played an important role in HF development.

Oar-miR-23b and oar-miR-133 can inhibit the expression of target genes and the translation of proteins, inhibit the proliferation and migration of SDFs, and promote the apoptosis of SDFs. The DFs in the connective tissue of sheep skin are derived from two different lineages. The cells in the upper dermis produce the dermal papilla and arrector pili muscles, the cells in the lower dermis produce the subcutaneous adipocyte layer, and the upper dermis is required for HF morphogenesis. DFs are cells associated with the formation of placodes in HF morphogenesis, which form dermal condensates and ultimately develop into dermal papilla and arrector pili muscles. Among them, the dermal papilla is the key “control center” in hair fiber growth and later circulation, and the arrector pili muscles are the key organs ensuring the normal growth of hair fibers. Oar-miR-23b and oar-miR-133 inhibit the proliferation and migration of SDFs, which may affect the development of placode and dermal condensate, leading to a disordered HF morphology and eventually leading to the abnormal development of HF in sheep.

## 5. Conclusions

In summary, oar miR-23b and oar miR-133 can inhibit the expression of *TGFβ2* and *NOTCH1* at the mRNA and protein levels in SDFs. Similarly, both oar-miR-23b and oar-miR-133 can inhibit the proliferation of SDFs and promote cell apoptosis while also affecting cell cycle progression and inhibiting the migration of SDFs. Both oar-miR-23b and oar-miR-133 regulate the expression of genes in the WNT, TGF-β, and Hippo signaling pathways. In short, oar-miR-23b and oar-miR-133 exert critical regulatory effects on SDFs, which, in turn, affects the occurrence of hair follicle morphology. Our research provides molecular markers and a molecular theoretical basis for revealing the genetic mechanism underlying ultrafine wool follicle development.

## Figures and Tables

**Figure 1 cells-13-00557-f001:**
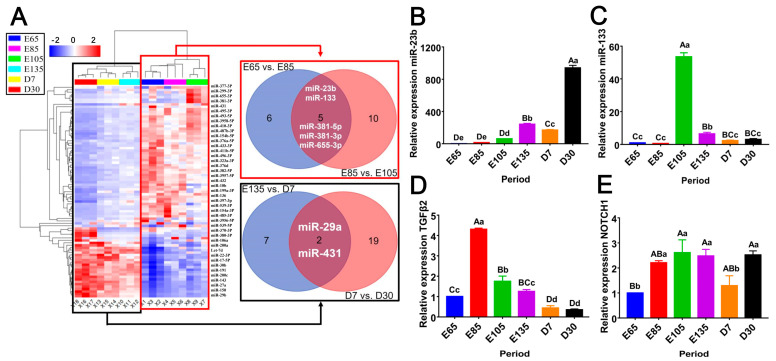
Clustering and expression analysis of miRNAs and target genes. (**A**) DE-miRNA heatmap analysis. Analysis of the gene expression of oar-miR-23b (**B**), oar-miR-133 (**C**), *TGFβ2* (**D**), and *NOTCH1*. (**E**) during HF development, (*n* = 3).

**Figure 2 cells-13-00557-f002:**
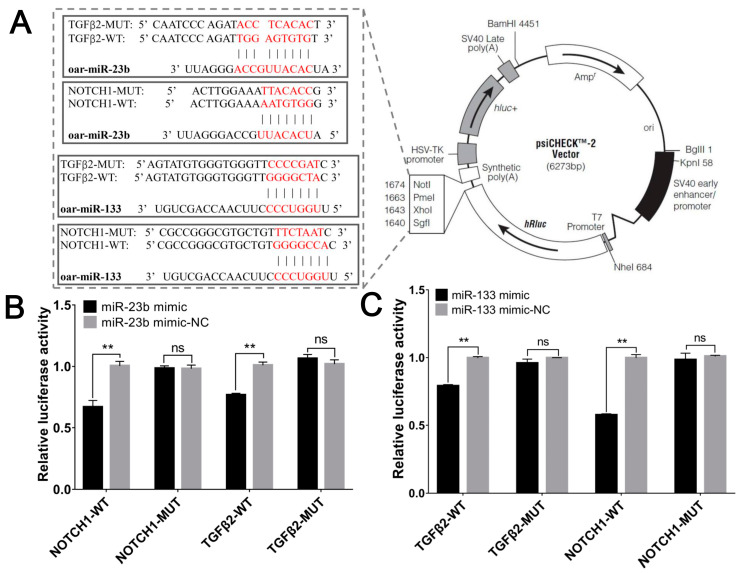
Oar-miR-23b and oar-miR-133 cotarget *NOTCH1*/*TGFβ2*. (**A**) The 3′UTR fragments of *NOTCH1* and *TGFβ2* containing the predicted WT and MUT target sites of oar-miR-23b/oar-miR-133 were cloned into the XhoI and NotI sites of the Renilla luciferase cassette in psiCHECK2. (**B**) Dual-luciferase reporter assay of the interaction of oar-miR-23b with its target genes (*n* = 3). (**C**) Dual-luciferase reporter assay of the interaction of oar-miR-133 with its target genes (*n* = 3), ** *p* < 0.01, ns *p* > 0.05.

**Figure 3 cells-13-00557-f003:**
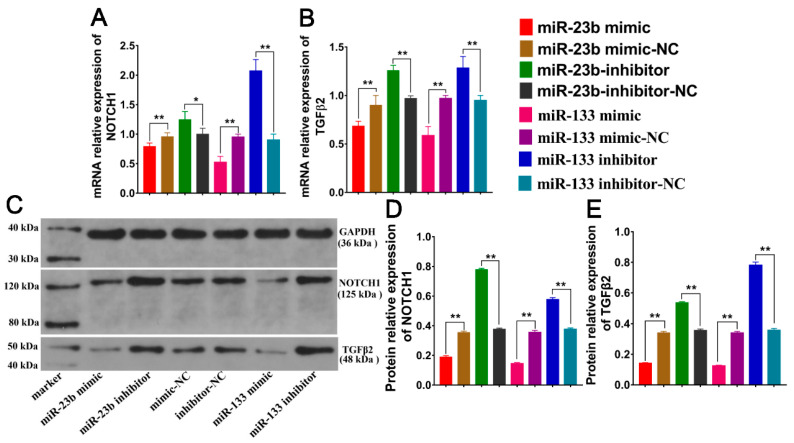
(**A**) Effects of oar-miR-23b and oar-miR-133 overexpression and inhibition on *NOTCH1* gene expression (*n* = 3). (**B**) Effects of oar-miR-23b and oar-miR-133 overexpression and inhibition on *TGFβ2* gene expression (*n* = 3). (**C**) Western blotting analysis of *NOTCH1* and *TGFβ2* expression after altering oar-miR-23b and oar-miR-133 expression. (**D**) Effects of oar-miR-23b and oar-miR-133 overexpression and inhibition on *NOTCH1*-based protein expression (*n* = 3). (**E**) Effects of oar-miR-23b and oar-miR-133 overexpression and inhibition on *TGFβ2* protein expression (*n* = 3), * *p* < 0.05, ** *p* < 0.01.

**Figure 4 cells-13-00557-f004:**
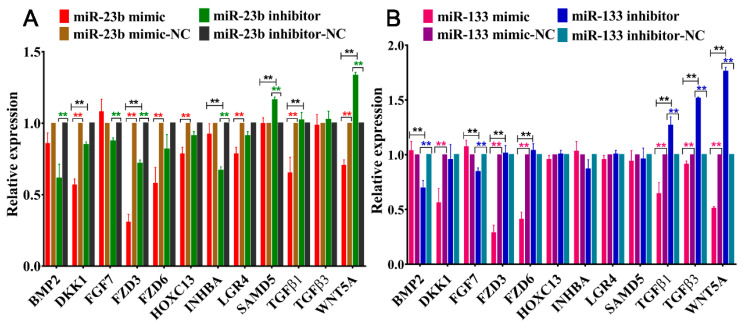
Effects of oar-miR-23b (**A**) and oar-miR-133 (**B**) on the expression of star genes in the PI3K-Akt, MAPK, TGF-β, Hippo, and WNT pathways (*n* = 3), ** *p* < 0.01.

**Figure 5 cells-13-00557-f005:**
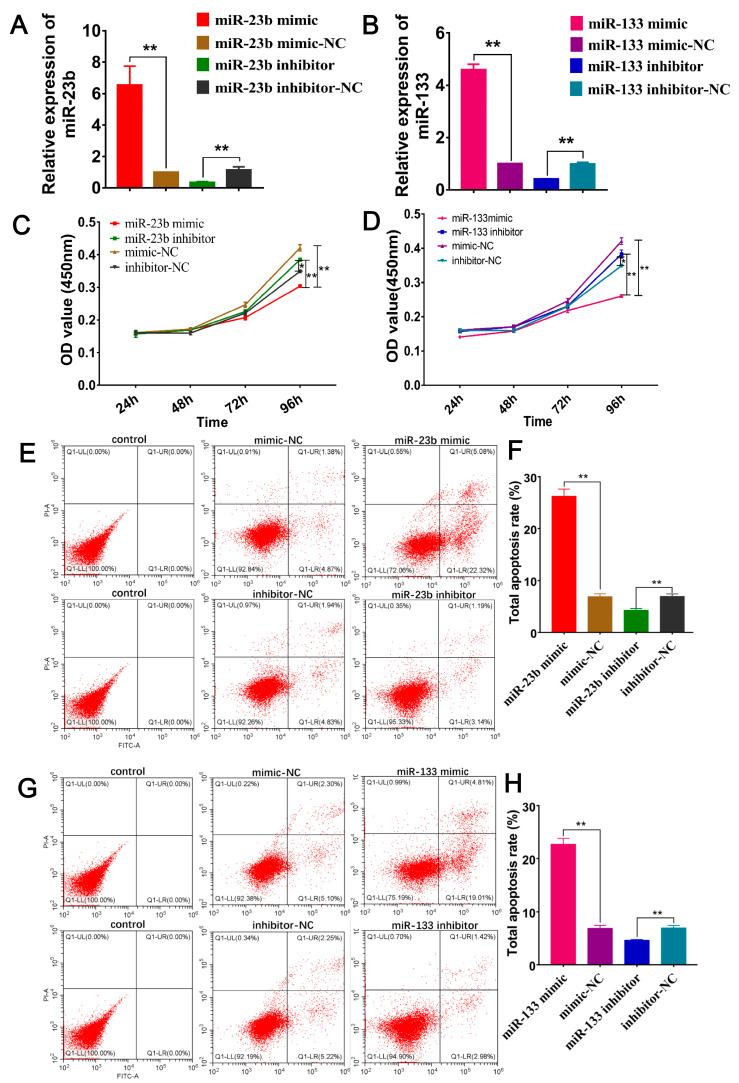
Effect of oar-miR-23b and oar-miR-133 on the proliferation and apoptosis of SDFs. (**A**,**B**) The transfection efficiency of oar-miR-23b and oar-miR-133 was verified by RT‒qPCR (*n* = 3). (**C**,**D**) CCK-8 cell proliferation assay (*n* = 6). (**E**,**G**) Apoptosis was analyzed by flow cytometry (*n* = 3). (**F**,**H**) Statistical analysis of the effects of oar-miR-23b and oar-miR-133 on apoptosis (*n* = 3), * *p* < 0.05, ** *p* < 0.01.

**Figure 6 cells-13-00557-f006:**
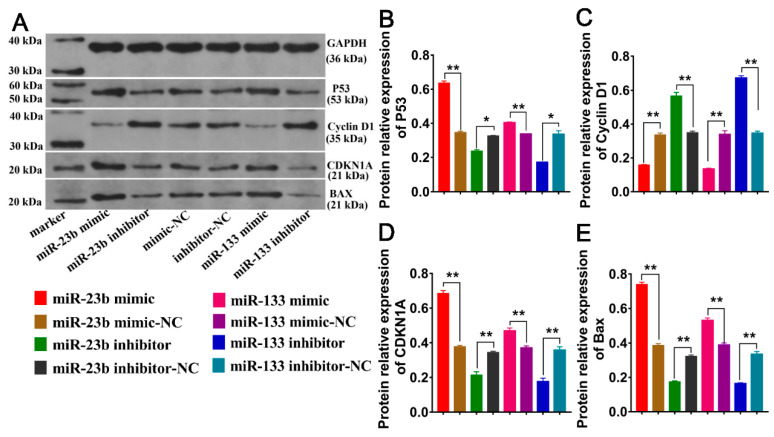
(**A**) Cell cycle-related protein expression was measured by Western blotting. (**B**) Relative protein expression of P53 (*n* = 3). (**C**) Relative protein expression of cyclin D1 (*n* = 3). (**D**) Relative protein expression of CDKN1A (*n* = 3). (**E**) Relative protein expression of BAX (*n* = 3), * *p* < 0.05, ** *p* < 0.01.

**Figure 7 cells-13-00557-f007:**
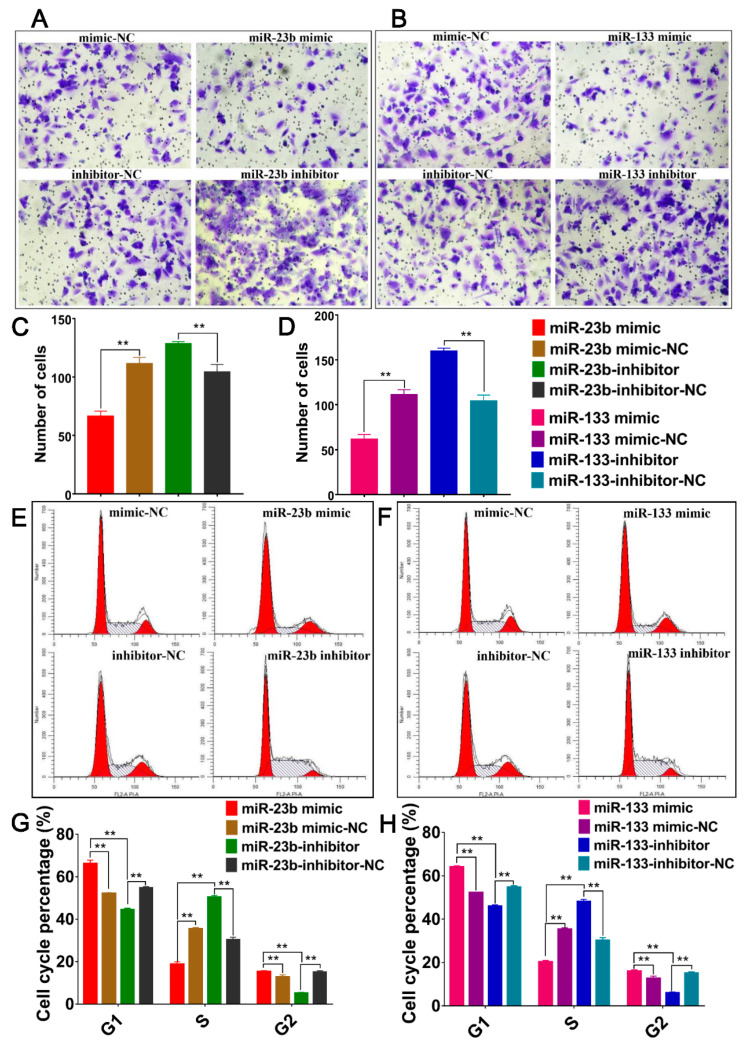
Effect of altering oar-miR-23b and oar-miR-133 expression on cell migration and cell cycle progression. (**A**,**B**) Image of cell migration, Magnification of 100 (*n* = 3). (**C**,**D**) Statistical analysis of cell migration (*n* = 3). (**E**,**F**) Effect of altering miRNA expression on cell cycle (*n* = 3). (**G**,**H**) Statistical analysis of cell cycle distribution (*n* = 3), ** *p* < 0.01.

## Data Availability

All the miRNA-seq data generated in this study were submitted to the NCBI SRA database under BioProject No. PRJNA705552 (https://www.ncbi.nlm.nih.gov/bioproject/?term=PRJNA705552, accessed on 15 February 2022). All the RNA-seq data generated in this study were submitted to the NCBI SRA database under BioProject No. PRJNA705554 (https://www.ncbi.nlm.nih.gov/bioproject/?term=PRJNA705554, accessed on 15 February 2022). We included other relevant data in this original manuscript file and/or in the supplementary information file. Nevertheless, the corresponding author will provide additional data related to these findings upon reasonable request.

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
