# Peer review of "MiR-23b and miR-133 Cotarget TGFβ2/NOTCH1 in Sheep Dermal Fibroblasts, Affecting Hair Follicle Development"

_cells, 2024, doi:10.3390/cells13060557_

Round 1

Reviewer 1 Report

Comments and Suggestions for Authors

Hair follicle development is a complex process involving communication between the epidermis and dermis and including multiple cell types.  Understanding this process is fundamental to a number of pathological conditions of the integument and, in the research presented here, to enhancing wool production in sheep.  The authors have previously identified several miRNAs that are differentially expressed at diverse stages of hair follicle development in sheep.  The present study was primarily focused on elucidating the functional roles of two of these miRNAs (oar-miR-23b and oar-miR-133) in isolated dermal fibroblasts.  These are interesting studies that help advance our understanding of these two miRs in dermal fibroblast gene expression and biological activity.  Several suggestions for the authors include:

-            - Portions of the data associated with Figure 1 have been previously presented by the authors in their prior publication cited here (Reference 48 – He et al., 2022).  Figures 1B-1D appear to be the same as Figure 1 of He et al., just plotted differently.  I would suggest elimination of  figure 1 as it is a bit of a distraction from the rest of the data, which is all focused on experiments in isolated fibroblasts.

-          -   It would be helpful to include 1-2 sentences at the beginning of the discussion of each new figure indicating the purpose of the experiments.  For instance, it would be helpful to have a sentence included at line 379 indicating “that vectors were created to evaluate interactions between oar-miR-23b and oar-miR-133 and target genes”. Then discuss the results.

-           -  It would be beneficial for the authors to include either a summary figure or a graphical abstract summarizing the data schematically.

Comments on the Quality of English Language

There are only minor grammatical errors detected.

Reviewer 2 Report

Comments and Suggestions for Authors

The study deals with the influence of miR23b and miR133 on dermal sheep fibroblasts and the development of hair follicles. As I come from a completely different field of research, it is difficult for me to grasp the relevance of the topic, so I can only describe it from a methodological point of view. In general, the methods and results were clearly presented. However, there are a few points that need to be adjusted before a possible publication.

- Were the samples tested for normal distribution and homogeneity? The statistical test to be used depends on this

- Which post hoc test was used?

- Information on the number of cases is generally missing

- From how many sheep were the isolated cells used? How was it ensured that the cells were fibroblasts? This information must be supplemented

- The number of cases and the statistics used must be stated in the figure legends.

Reviewer 3 Report

Comments and Suggestions for Authors

This paper strongly addressed the effects of miR-23b and miR-133 on wool hair follicle development using sheep dermal fibroblasts. Although the topic is interesting and the authors provided sufficient data to substantiate this hypothesis, the following issues must be resolved prior to publication.

1. In this study, the authors used sheep dermal fibroblasts to determine the effects of miR-23b and miR-133 on hair follicle development and regulation. However, generally, “dermal papilla cells” are used rather than “dermal fibroblasts” to study hair formation. Therefore, authors need to clearly state why they used “dermal fibroblasts” in the Introduction or Discussion section. If it is due to limitations in their resources, they need to describe in the Discussion section that their follow-up studies will investigate the effects of miR-23b and miR-133 on sheep dermal papilla cells.

2. The authors thoroughly investigated the effects of miR-23b and miR-133 on various signaling pathways in this paper. Previous studies have shown that hair-inducing activity is closely related to multiple signaling pathways. To ensure readers' broad understanding, the authors need to add detailed explanations by referring to papers such as Rendl et al., Genes Dev., 2008, 543-557; Lee et al., PLoS One, 2012, e34152 in the Introduction or Discussion section of the revised manuscript.

3. Although the author wrote the manuscript in good quality English, minor errors were found throughout the paper. Careful checking is required during revision, including the examples below.

1) The authors often used both the abbreviated and full forms simultaneously. For example, they used the abbreviated term "min" throughout the paper, but also used "minutes" on two occasions (2.1. Animal selection and skin tissue preparation, Page 4-Lane 155; 2.11. Flow cytometry analysis, Page 6-Lane 298). The authors should use terminology consistently in the revised manuscript.

2) As the authors well know, the 'c' in 'β-catenin' is capitalized only when used at the beginning of a sentence. Therefore, 'β-Catenin' on Page 2, Lane 83 should be changed to 'β-catenin'. Also, 'b-catenin-deficient' on Page 16, Lane 569 must be appropriately modified to 'β-catenin-deficient'.

Round 2

Reviewer 3 Report

Comments and Suggestions for Authors

[70] is not associated with Wnt signaling or hair-inducing activity. Please remove this article from the reference list of the revised manuscript. Aside from this issue, the authors have sufficiently addressed all the issues I raised.

Author Response

Responds to the Reviewer 3:

1. Response to comment: [70] is not associated with Wnt signaling or hair-inducing activity. Please remove this article from the reference list of the revised manuscript. Aside from this issue, the authors have sufficiently addressed all the issues I raised.

Response: We deeply apologize for any unclear content in the manuscript and thank you very much for your suggestion. We have removed the reference [70] from the manuscript.